# Endotoxemia in Acute Heart Failure and Cardiogenic Shock: Evidence, Mechanisms and Therapeutic Options

**DOI:** 10.3390/jcm12072579

**Published:** 2023-03-29

**Authors:** Maxime Nguyen, Thomas Gautier, David Masson, Belaid Bouhemad, Pierre-Grégoire Guinot

**Affiliations:** 1Department of Anesthesiology and Intensive Care, Dijon University Hospital, F-21000 Dijon, Franceguinotpierregregoire@gmail.com (P.-G.G.); 2Université Bourgogne-Franche Comté, UMR1231, F-21000 Dijon, France; 3INSERM, UMR1231, F-21000 Dijon, France; 4LipSTIC LabEx, F-21000 Dijon, France

**Keywords:** endotoxemia, lipopolysaccharide, acute heart failure, cardiogenic shock, translocation

## Abstract

Acute heart failure and cardiogenic shock are frequently occurring and deadly conditions. In patients with those conditions, endotoxemia related to gut injury and gut barrier dysfunction is usually described as a driver of organ dysfunction. Because endotoxemia might reciprocally alter cardiac function, this phenomenon has been suggested as a potent vicious cycle that worsens organ perfusion and leading to adverse outcomes. Yet, evidence beyond this phenomenon might be overlooked, and mechanisms are not fully understood. Subsequently, even though therapeutics available to reduce endotoxin load, there are no indications to treat endotoxemia during acute heart failure and cardiogenic shock. In this review, we first explore the evidence regarding endotoxemia in acute heart failure and cardiogenic shock. Then, we describe the main treatments for endotoxemia in the acute setting, and we present the challenges that remain before personalized treatments against endotoxemia can be used in patients with acute heart failure and cardiogenic shock.

## 1. Introduction

Acute heart failure with cardiogenic shock is a frequently occurring and severe condition that combines low cardiac output, altered blood pressure and organ hypoperfusion. In cardiovascular diseases, the cross-talk that occurs between the gut and the heart is a subject of interest, and there is growing evidence regarding the link between gut microbiome diversity, endotoxemia and systemic inflammation in chronic heart failure (CHF) [1]. During cardiogenic shock, reduced cardiac output results in inadequate oxygen delivery to tissues and organs, including the digestive tract. Physiologically, the surface of the gut is an area that absorbs nutrients while acting as a barrier, preventing noxious substances and bacteria originating from the gastro-intestinal lumen, including lipopolysaccharides (LPS, endotoxins), from reaching the bloodstream. Gut ischemia-reperfusion is a condition that promotes endotoxemia (i.e., the presence of LPS in the circulation) through gut barrier failure [2]. Because endotoxemia reduces cardiac performance [3], it can lead to a vicious cycle including reduced cardiac output [4,5], gut ischemia/hypoperfusion, translocation of endotoxins, inflammation and organ failure. However, unlike vascular surgery or mesenteric ischemia, there is no complete interruption of the digestive arterial blood flow during acute heart failure (AHF) [6]. Consequently, the pathogenesis of gut barrier injury and endotoxemia might differ. In addition, because LPS-induced cardiac dysfunction has been demonstrated to be dose-dependent in animal models [7], the extrapolation of one disease to another might not be straightforward.

It is currently accepted that endotoxemia may play a role in the spiral of multiple organ dysfunction with systemic inflammation and microcirculatory alterations during AHF. However, no review has specifically focused on the evidence and the role of endotoxemia during AHF and cardiogenic shock. Moreover, despite available options, there are no guidelines for therapy targeting endotoxemia in this indication.

In this work, we review the data supporting endotoxemia as a pathological mechanism of inflammation and multiple organ failure in humans with AHF and cardiogenic shock. Our objective is to identify the next steps required before implementing treatment targeted to endotoxemia in those patients.

## 2. The Concept: Gut-Derived Lipopolysaccharides Promote Inflammation in AHF

### 2.1. What Are LPS and How Do We Measure Them?

LPS are a component of the outer membrane of gram-negative bacteria. They are known to activate host immunity and are probably the most studied pathogen-associated molecular patterns. LPS exert their noxious effect primarily by triggering inflammation via the TLR-4 pathway [8], which has been suggested to be one of the triggers of systemic inflammation and multiple organ failure during critical illness [9]. More specifically, after TLR-4 activation, LPS activate pro-inflammatory pathways, resulting in endothelial dysfunction, coagulopathy, and cardiovascular dysfunction [10,11]. In infection settings, the activation of the immune response represents a host defense against the pathogen. In parallel, this inflammatory response is balanced by an anti-inflammatory response. When this host response is dysregulated, patients developed organ dysfunctions that are defined as sepsis [12]. It is also possible that the dysregulation of the immune response might be due to a disproportionate pro-inflammatory response, resulting in acute early multi-organ failure [13], or by exaggerated T-cell dysregulation, which is responsible for immunosuppression and results in secondary nosocomial infection [14].

In physiology, gram negative bacteria are part of the microbiome contained within the digestive tract. Depending on the bacterial species, the LPS derived might exert different levels of pathogenicity [15]. Therefore, it has been hypothesized that a shift in gut microbiota could have a major influence upon the consequences of LPS translocation [16].

There are two methods that can be used to directly reveal endotoxemia: measuring endotoxin activity (presence of biologically active LPS in blood) or determining endotoxin mass (quantification of LPS molecules, active and inactive). Because LPS exist in both active and inactive forms in the blood, endotoxin activity might be low despite the presence of genuine endotoxemia (presence of inactive LPS). Thus, a quantitative measurement of LPS seems mandatory to assess the translocation and elimination process, while a combination of quantitative and qualitative methods makes it possible to assess an individual’s ability to neutralize endotoxins [17]. Most, if not all, clinical studies only rely on measurements of endotoxin activity to assess endotoxemia, which can result in very low or even negative results. This might explain why endotoxemia is often overlooked in conditions associated with impaired gut barrier.

### 2.2. Gut Barrier Function, Heart Failure, Ischemia-Reperfusion and LPS Translocation

In humans, several mechanisms prevent LPS in the gut from reaching the blood flow [18]. For instance, intra-luminal enzymes, proteins and the mucus layer provide an in situ protective mechanism against LPS translocation. The intestinal epithelium and tight junctions also prevent LPS translocation. Nevertheless, despite these mechanisms and even in the absence of acute gastro-intestinal injury, low grade endotoxemia can occur [19]. In this situation, because LPS are too large to transit through the paracellular space, a transcellular pathway is likely. Indeed, LPS from the gut lumen might cross the intestinal epithelial layer through endocytosis, followed by exocytosis at the basal enterocyte pole by goblet cell-associated antigen passage [20]. Otherwise, LPS might be absorbed similar to dietary fatty acid via chylomicron synthesis [21,22]. Once they have entered the portal circulation, part of LPS are transferred to lipoproteins by the action of the phospholipid transfer protein (PLTP). LPS bound to lipoproteins are inactivated (it cannot riggers TLR-4 response) [23]. LPS transfer and binding to lipoproteins also promote hepatic LPS elimination. Indeed, the liver represents an important barrier to prevent gut-derived LPS entry into the systemic circulation by exerting a first hepatic pass effect [24].

Acute systemic aggressions might increase paracellular permeability and have been correlated with endotoxemia [25,26,27]. Ischemia-reperfusion is a particular kind of aggression that damages the cells and tissues [28]. Gut mucosa is injured during both the ischemic phase and after reperfusion [29], and gut barrier function might be impaired, thus promoting translocation of gut content and inflammation [18,29]. In patients with AHF, low cardiac output, regional vasoconstriction and congestion might alter oxygen delivery (i.e., non-occlusive ischemia) [30,31]. Reciprocally, endotoxemia impairs cardiac function [3] and mucosal gut oxygenation [32], which might create a vicious cycle (Figure 1). Reperfusion eventually occurs once the patient has been successfully resuscitated. Interestingly, it has been reported that ischemia-reperfusion of the lower limbs may alter gut structure and permeability, resulting in potential gut barrier dysfunction even though the primary injury does not affect the digestive tract [33]. This emphasizes the role of inflammatory mediators in gut permeability.

Visceral ischemia reperfusion caused by damaging gut mucosa and enterocytes [34] increases permeability, including for large molecules [27], and promotes endotoxemia. Nevertheless, whether endotoxemia is promoted by increased transcellular transport, abnormal paracellular permeability or both is not yet elucidated. Indeed, while it has been observed that regional ischemia and reperfusion of the gut could induce simultaneous tight junction loss and endotoxemia [35] (strongly suggesting that increased paracellular permeability is one of the causes of LPS translocation), those simultaneous observations do not imply causality, and there is no direct proof regarding paracellular transport of LPS after ischemia-reperfusion in the literature [20]. After translocation, a protective mechanism occurs: LPS might be transferred to lipoproteins, promoting their inactivation and elimination [23]. For LPS originating from the splanchnic area, binding to lipoproteins has been demonstrated to promote a first hepatic pass effect, and part of the LPS burden might be eliminated by the liver before reaching the systemic circulation [24].

## 3. The Evidence: Clinical Findings

### 3.1. Cardiogenic Shock and Gut Injury

In critically ill patients, the two most studied biomarkers of gut barrier dysfunction are the intestinal fatty acid binding protein (I-FABP), which is a marker of enterocyte suffering, and citrulline, which reflects enterocyte mass [36]. A few teams have investigated I-FABP measurements in the context of cardiogenic shock. During cardiogenic shock, I-FABP was reported to be higher at ICU admission in non-survivors than in survivors [37]. Although we did not find any comparison between patients with and without cardiogenic shock, higher I-FABP has been reported in patients with shock and patients requiring catecholamine [38,39]. Results suggest that I-FABP might not reflect gut barrier permeability. Indeed, in the context of mild systemic aggression, it was demonstrated that I-FABP could be dissociated from intestinal permeability in healthy patients with systemic inflammatory response syndrome (SIRS) induced by experimental endotoxemia, suggesting that gut barrier dysfunction may occur in the absence of enterocyte necrosis [40]. Similarly, in patients undergoing cardiac surgery with cardiopulmonary bypass, there was no correlation between I-FABP and LPS measured by mass spectrometry [41]. On the contrary, in patients successfully resuscitated from cardiac arrest, higher endotoxin activity was associated with higher I-FAPB [42], which suggests that the association between I-FABP and LPS translocation depends on the intensity and/or nature of the injury. No study has specifically focused on citrulline in AHF or cardiogenic shock. Despite being largely used for research purposes, those biomarkers are not available at bedside. More available biomarkers, such as ammonia, have been proposed as markers of abdominal injury in patients with heart failure [43]. Nevertheless, their implementation into daily practice probably requires further investigation.

### 3.2. Acute Heart Failure and Endotoxemia

Several studies focused on endotoxemia during AHF in the absence of shock. AHF is a clinical entity that regroups different phenotypes (acute on chronic/de novo, ischemic/other etiology, preserved/altered left ventricular function, congestive/non-congestive) that might impact endotoxemia. An observational study suggested that myocardial infarction is associated with increased levels of endotoxins in relation to alterations of intestinal permeability. Indeed, patients with myocardial infarction had higher serum level of LPS and zonulin (a protein that regulates intestinal barrier function [44] and could be used to assess gut permeability [45]) than healthy centenarians [46]. Zhou et al. provide additional information by demonstrating that myocardial infarction may promote gut barrier dysfunction and endotoxemia with LPS activity increased by 2–3 fold on day 1 [47]. Moreover, the authors reported a link between gut barrier dysfunction, endotoxemia and further cardiovascular events. Decompensation of chronic heart failure might also promote endotoxemia. Indeed, endotoxin activity was reported to be higher in patients with decompensated/congestive heart failure compared to both stable patients with chronic heart failure and healthy patients [48,49]. In patients with decompensated heart failure, a gradient in endotoxin activity between the hepatic vein and the left ventricle suggests a splanchnic origin, i.e., the digestive translocation of LPS [49]. Interestingly, decongestion has been associated with lowering plasma endotoxin activity in patients with congestive heart failure in two studies [48,50].

### 3.3. Cardiogenic Shock and Endotoxemia

We found four articles studying endotoxemia in cardiogenic shock (Table 1). The first article, by reporting elevated biomarkers of inflammation (CRP, PCT, cytokines) and pyrexia in patients without bacteremia provides indirect proof of endotoxemia (as endotoxin was not directly measured) [51]. In the second report, Ramirez et al. reported low titers of IgM EndoCAb (also an indirect measurement of endotoxemia) in 22 patients with cardiogenic shock, leading to a hypothesis of endotoxin exposure. Indeed, endotoxemia is not measured directly by this method, but is suggested by the drop of endotoxin antibodies in patients’ plasma, and the underlying hypothesis of antibody consumption is due to endotoxin exposure. However, the actual endotoxins were not measured, and the antibody titer, which provides only indirect assessment of endotoxemia, was not associated with mortality [52]. In the third article, endotoxin activity was measured in 16 patients with cardiogenic shock, but only one of the patients was found to have high endotoxin activity [53]. The fourth study reported endotoxin activity in 39 patients under venoarterial extracorporeal membrane oxygenation (VA-ECMO). In these patients (16 cardiac arrest, 17 heart failure, 4 post cardiotomy and 2 septic shock), the incidence of endotoxemia was only 9% within the first 24 h and 5% between 24 and 48 h. There was no difference in endotoxin activity between survivors and non-survivors [54].

Altogether, we found no direct proof of endotoxemia in patients with cardiogenic shock in the literature, and no study reported quantitative measurement (i.e., mass) of endotoxins in the circulation. The few studies that assessed endotoxemia activity suggested a low incidence of endotoxemia in this population, but the correlation between LPS mass and activity was weak [55]. Thus, low endotoxin activity cannot rule out endotoxemia.

## 4. Treatment Perspectives for Endotoxemia in Acute Heart Failure

### 4.1. The Hemodynamic Consequences of Heart Failure

Because the consequences of AHF (ischemia-reperfusion injury and tissue hypoperfusion) are involved in low systemic blood flow and are triggers of SIRS and endotoxemia, the first therapeutic goal may be to restore blood flow, oxygen delivery and tissue perfusion. Once perfusion is restored, treating venous congestion may help to reduce the endotoxin burden [48]. In patients with ischemic cardiogenic shock/heart failure, early vascularization of the culprit lesion should be considered [56,57].

### 4.2. Specific Treatments for Endotoxemia

In patients with cardiogenic shock, the excessive triggering of inflammation by pathogen-associated molecular patterns (PAMPs) and damage-associated molecular patterns (DAMPs) is one of the mechanisms leading to organ failure and other adverse outcomes^8^. Because endotoxins can trigger inflammation, it seems logical that reducing LPS and TLR-4 activation could reduce inflammation and its harmful consequences in the context of digestive LPS translocation. Although there are no clear indications for any therapy directly targeting endotoxemia, some strategies have been developed to counter the consequences of endotoxemia. We have classified these interventions into the four groups described below (Figure 1).

The first type of intervention aims to reduce endotoxin load and/or promote gut barrier integrity directly in the digestive tract. Selective digestive decontamination, prebiotic, probiotic administration and fecal transplantation have been described as means of modulating the gut microbiota [58,59] Gut microbiota has been associated with inflammation in patients with heart failure [60]. Because there are several type of LPS exerting several pathogeneses, modulating the gut microbiota might be an interesting strategy that could potentially reduce the intra-luminal load of LPS but also modify LPS type. Nevertheless, the effects of selective decontamination on endotoxemia in cardiac surgical patients are controversial [61,62]. One explanation might be given by the GutHeart trial that reports, in the context of chronic heart failure, that two microbiome-based interventions (probiotics and antibiotics) were inefficient to change the microbiota diversity [63]. Enteral nutrition might also prevent endotoxemia by promoting gut integrity. For instance, in critically ill patients, enteral nutrition might prevent mucosa atrophy and help restore enterocyte mass [64]. Enteral nutrition has also been described as a way to maintain the intestinal alkaline phosphatase activity that is able to phosphorylate LPS [65].

Another type of intervention is extracorporeal LPS removal. Many extracorporeal adsorption devices have been developed over the years [66]. Usually, the membranes are functionalized with a molecule that allows charge interactions, thereby adsorbing LPS. Among them, Polymyxin B is the most studied and has been demonstrated to eliminate LPS in vitro [67]. Nevertheless, when polymyxin B was tested in large randomized controlled trials in the context of septic shock, including in populations with high endotoxin activity, it failed to improve outcomes [68,69,70]. Membrane functionalized with acrylonitrile and methanesulfonate have also been demonstrated to adsorb LPS [67]. Nevertheless, there are no large randomized control trials (RCT) evaluating LPS adsorption with such a device. In a small-sample study upon patients with shock septic and endotoxemia, this device was reported to reduce endotoxin activity at early time points [71]. Nevertheless, extracorporeal adsorption in the context of cardiogenic shock and septic shock are different. Indeed, during sepsis, inflammation is a key factor for fighting the pathogen, so the removal of cytokines is more likely to be deleterious. Indeed, two RCTs that focused on plasma filtration and adsorption of cytokines in septic shock patients were interrupted prematurely due to possible harmful effects [72,73]. Furthermore, the endotoxin load in the context of cardiogenic shock is probably much lower than in gram-negative sepsis, and LPS is probably not the main trigger of inflammation. The utility of LPS extracorporeal removal in this indication is currently under investigation [74].

It is also possible to directly target the LPS-TLR-4axis by anti-LPS [75,76] treatment, TLR-blockade [77,78] or anti-cytokine treatment [79,80]. To date, large clinical trials have failed to demonstrate any benefit for those molecules in acute settings. The main endogenous pathway for LPS elimination is reverse lipopolysaccharide transport (RLT). This pathway involves the transfer of LPS by the phospholipid transfer protein (PLTP) to lipoproteins (LDL, HDL) that neutralize LPS and promote their elimination through the hepatobiliary route [81]. The fourth type of intervention aims to enhance this pathway, and several strategies are under investigation. Firstly, the administration of recombinant HDL is one way to increase the pool of lipoproteins involved in LPS inactivation and transport to the liver. This strategy has been demonstrated to improve survival in animal models of sepsis [82], and to decrease inflammation in healthy volunteers [83]. Secondly, since LDL are also involved in LPS binding and transport, they might be an interesting target for enhancing RLT. On the same topic, two studies demonstrated the relevance of inhibiting PCSK9 to promote the clearance of LDL-bound LPS (by increasing the expression of the LDL receptor) [84,85]. Thirdly, intravenous administration of a phospholipid emulsion with the aim of enhancing the ability of lipoproteins to bind to LPS has also been proposed. However, this strategy has failed to demonstrate any benefit in patients with gram-negative sepsis [86]. Fourthly, increasing plasma phospholipid transfer protein (PLTP) activity might also enhance the binding of LPS to circulating lipoproteins. Accordingly, the administration of recombinant PLTP has been demonstrated to improve survival in animals models of endotoxemia [87]. Because of the particular vascularization of the gut, LPS undergo a first hepatic elimination before reaching the systemic compartment [24], and these therapies might provide early LPS inactivation and clearance in the context of gut-derived endotoxemia (Figure 1). Nevertheless, these strategies have not been validated in human patients, and because lipoprotein metabolism and lipid transfer activity are modified during critical illness [88], tailoring these interventions to the peculiar lipoprotein profiles might be necessary.

## 5. Future Directions and Therapeutic Options

During cardiogenic shock, several challenges must be overcome before strategies can be implemented to treat endotoxemia (Table 2).

First, because multiple mechanisms are responsible for adverse disease progression [5,89], endotoxemia might only be a significant trigger of inflammation in selected patients with cardiogenic shock. In addition, the equilibrium between the pro- and anti-inflammatory response is a major concern in critically ill patients, and the immunological response to aggression is highly variable between individuals [13]. Therefore, the “one-size-fits-all” concept applies poorly to immunology, and personalizing treatments will probably be necessary [90]. Altogether, identifying patients with significant endotoxemia and a dysregulated pro-inflammatory immune response would be an important prerequisite to implement those treatments. Second, our understanding of the pathological mechanisms underlying LPS translocation in the injured gut is still incomplete. The precise and exhaustive identification of the protective mechanisms of the epithelial barrier must enable the development of strategies directly aiming to reduce LPS translocation at very early time points, i.e., before LPS reaches the portal blood flow. Finally, the different strategies for LPS detoxification presented above should be tailored and evaluated by randomized controlled trials.

## 6. Conclusions

The role of endotoxemia in the genesis of cardiogenic shock has not yet been fully explored. While in heart failure, endotoxin activity seems to increase with disease progression and congestion, in the few studies that directly explored endotoxemia in cardiogenic shock, the incidence of endotoxemia was low. However, because only endotoxin activity (and not quantity) is reported, the LPS burden in patients with cardiogenic shock might be underestimated.

While therapeutics directed to endotoxemia are already available, this work underlines that further research is needed before implementing such treatment. In particular, identifying patients with associated cardiogenic shock and significant endotoxemia seems to be the first important step. The development of biomarkers validated against quantitative LPS measurements might be an efficient way to identify this population.

Contribution to the Field Statement: In cardiogenic shock, endotoxemia is described as a driver of organ failure. Treatments targeted to endotoxemia are available. Here, we review the evidence beyond endotoxemia in cardiogenic shock with the aim to implement such treatment. We found that, despite the admitted paradigm of gut-derived endotoxemia, the reported incidence of endotoxemia in cardiogenic shock appeared to be low. Thus, it is unlikely that targeting endotoxemia in unselected patients with cardiogenic shock would improve outcome. In consequence, a personalized medicine approach seems more appropriate, and future research should focus on identifying patients with cardiogenic shock and significant endotoxemia.

## Figures and Tables

**Figure 1 jcm-12-02579-f001:**
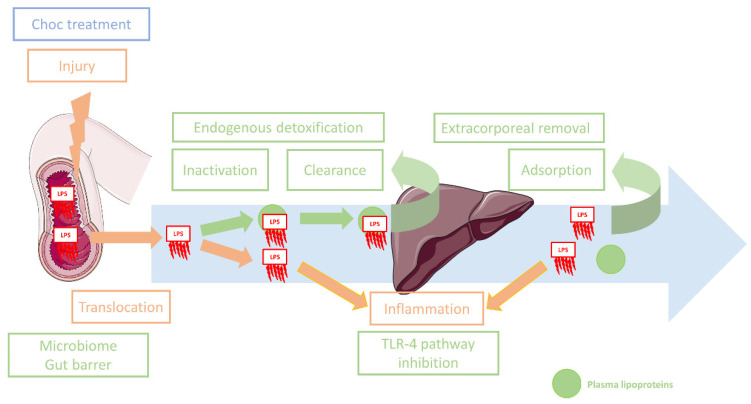
Lipopolysaccharide gut translocation, mechanisms and possible treatments. Noxious mechanisms are represented in orange. Potential targeted therapeutic interventions are represented in green. Symptomatic therapeutic interventions are represented in blue. LPS: Lipopolysaccharides.

**Table 1 jcm-12-02579-t001:** Studies reporting endotoxemia in cardiogenic shock by endotoxin measurement method.

	Method	Ref	Incidence
Quantitative measurement	-	None	-
Activity measurement	EAA	[53]	Low (6%)
	EAA	[54]	Low (9%)
Indirect proof	CRP, PCT, Cytokine and negative blood culture	[51]	-
	IgM EndoCAb	[52]	-

EAA: Endotoxin activity assay; C-reactive protein, PCT: procalcitonin. EndoCAb: Core antibodies directed to endotoxin.

**Table 2 jcm-12-02579-t002:** Main research needed before implementing therapy targeted to endotoxemia in patients with acute heart failure and/or cardiogenic shock according to the authors.

Thematic	Objective	Type of Research
Mechanism of translocation	Determining the route for endotoxin translocation in patients with ischemia-reperfusion injury	Experimental
Patient selection	Determining a phenotype/group of patients that would benefit from endotoxin removal	Observational cohort
Detoxification	Evaluating the different therapeutic targeted to endotoxemia available (in the population of interest).	Randomized controlled trial

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
