# Peer review of "Endotoxemia in Acute Heart Failure and Cardiogenic Shock: Evidence, Mechanisms and Therapeutic Options"

_jcm, 2023, doi:10.3390/jcm12072579_

Round 1

Reviewer 1 Report

I would like to congratulate the authors for this highly interesting and relevant review on endotoxemia in heart failure and cardiogenic shock entitled: “Endotoxemia in acute heart failure and cardiogenic shock: evidence, mechanisms and therapeutic options”. The review is original and comprehensive, giving insight to a complicated theme that is not completely understood.

Addressing the following points will provide additional value to the manuscript and better understanding of this theme.

Minor Comment:

-       The evidence: clinical findings paragraph. Acute heart failure and endotoxemia subparagraph. 

I would add a small paragraph on “everyday practice” measurable markers of of abdominal injury in heart failure patients, especially highlighting those that appear to have a clinical and prognostic role. See the manuscript of Frea S. et al. Clinical and prognostic role of ammonia in advanced decompensated heart failure. The cardio-abdominal syndrome? Int J Cardiol. 2015 Sep 15;195:53-60. doi: 10.1016/j.ijcard.2015.05.061. Epub 2015 May 12. PMID: 26022800.

Author Response

Addressing the following points will provide additional value to the manuscript and better understanding of this theme.»

Response: We would like to thank the reviewer for his positive feedback on our work.

Minor suggestions

-       The evidence: clinical findings paragraph. Acute heart failure and endotoxemia subparagraph. 

I would add a small paragraph on “everyday practice” measurable markers of of abdominal injury in heart failure patients, especially highlighting those that appear to have a clinical and prognostic role. See the manuscript of Frea S. et al. Clinical and prognostic role of ammonia in advanced decompensated heart failure. The cardio-abdominal syndrome? Int J Cardiol. 2015 Sep 15;195:53-60. doi: 10.1016/j.ijcard.2015.05.061. Epub 2015 May 12. PMID: 26022800.

Response: We thanks the reviewer for this comment. A sentence about the use of biomarker in everyday practice has been added in the “cardiogenic shock and gut injury” paragraph. In particular the fact that daily available biomarkers such as ammonia have been studied is now discussed.

Reviewer 2 Report

Nguyen et al., review the data supporting endotoxemia as a pathological mechanism of inflammation and multiple organ failure in humans with acute heart failure and cardiogenic shock. The authors also described the main treatments for endotoxemia in the acute setting and the challenges to be addressed.

Overall the review is nicely written and addresses some of the key issues we are facing in deciding the personalized treatment options. I only have few comments.

1. A flowchart with treatment schema for patients with CS and AHF highlighting the causes and concerns will definitely help readers to get the message clearly.

2. Does discussing SHOCK trial or CULPRIT-SHOCK trial will be helpful to get some more insights?

Author Response

 “ Nguyen et al., review the data supporting endotoxemia as a pathological mechanism of inflammation and multiple organ failure in humans with acute heart failure and cardiogenic shock. The authors also described the main treatments for endotoxemia in the acute setting and the challenges to be addressed.

Overall the review is nicely written and addresses some of the key issues we are facing in deciding the personalized treatment options. I only have few comments.

  1. A flowchart with treatment schema for patients with CS and AHF highlighting the causes and concerns will definitely help readers to get the message clearly.

Response: We thank the reviewer for the comment. Medical treatment of AHF and cardiogenic shock are well studied and widely published in several review/guidelines. But they are not part of the present review that focus on endotoxemia. To date we do not have recommended treatment for endotoxemia in cardiogenic shock. Thus, we believe that adding a flow chart highlighting the causes and concerns of cardiogenic shock will not add value on the topic.

  1. Does discussing SHOCK trial or CULPRIT-SHOCK trial will be helpful to get some more insights?”

Response: We would like to thank the reviewer for his/her positive feedback on our work

We apologize for the lack of clarity. The fact that patients with ischemic cardiogenic shock/ heart failure should undergo early revascularization of the culprit lesions has been added into the first paragraph of treatment perspectives.

Reviewer 3 Report

The authors conducted an interesting review. However, several aspects need to be addressed before publication:

The authors mention different methods of LPS analysis -> Please explain/describe

Discuss the role of dysbiosis – shift of microbiome as a source of LPS

Please discuss the role of the liver as an important barrier before LPS enters systemic circulation (LPS clearance?)

Role of direct immune response / priming directly at the site of entry (mucosal immune system) is not covered by the authors.

Role of lipids is not covered.

The authors must discuss the role of LPS that has entered the circulation – especially in these acute settings -> immune activation / senescence / exhaustion.

The authors should consider and discuss the recent GutHeart trial.

Author Response

 “ The authors conducted an interesting review. However, several aspects need to be addressed before publication:

The authors mention different methods of LPS analysis -> Please explain/describe

Discuss the role of dysbiosis – shift of microbiome as a source of LPS

Please discuss the role of the liver as an important barrier before LPS enters systemic circulation (LPS clearance?)

Role of direct immune response / priming directly at the site of entry (mucosal immune system) is not covered by the authors.

Role of lipids is not covered.

The authors must discuss the role of LPS that has entered the circulation – especially in these acute settings -> immune activation / senescence / exhaustion.

The authors should consider and discuss the recent GutHeart trial.”

Response: We would like to thank the reviewer for his/her positive feedback on our work

We apologize for the lack of clarity.

1/ The difference between different methods of LPS analysis has been described in a separate paragraph in the “What are LPS and how do we measure them section”.

2/ The impact of role dysbiosis has been discussed into the “What are LPS and how do we measure them section”.

3/ The first hepatic pass effect has been discussed in the “Gut barrier function, heart failure, ischemia-reperfusion and LPS translocation” section

4/ We choose not to cover the role of direct immune response as we wanted to focus on gut translocation and consequences. Therefore, we felt that interaction between gut microbioma was out of the scope and gut mucosa would complexify the manuscript.

5/ We thank the reviewer for his comment. The role of lipids in LPS inactivation/ elimination has been discussed in the “Gut barrier function, heart failure, ischemia-reperfusion and LPS translocation” section.

6/ The immune activation/ and exhaustion is now discussed in the “what are LPS and how do we measure them” section

7/ The GutHeart trial is now discussed in the “specific treatment for endotoxemia” section